∂ | **Open Peer Review** | Antimicrobial Chemotherapy | Research Article

# *fosA11*, a novel chromosomal-encoded fosfomycin resistance gene identified in *Providencia rettgeri*

Wei Lu,[1,2] Shihan Zhou,[2] Xueli Ma,[1] Nuo Xu,[2] Dongxin Liu,[1] Keqing Zhang,[1] Yongke Zheng,[2,3] Shenghai Wu[1,2]

**ABSTRACT**  This study investigated resistance genes corresponding to the fosfomycin resistance phenotype in clinical isolate *Providencia rettgeri* W986, as well as characterizing the enzymatic activity of FosA11 and the genetic environment. Antimicrobial susceptibility testing was performed using the agar microdilution method based on the Clinical and Laboratory Standards Institute guidelines. The whole genomic sequence of *Providencia rettgeri* W986 was obtained using Illumina sequencing and the PacBio platform. The *fosA-11* gene was amplified by PCR and cloned into the pUCP20 vector. The recombinant strain pCold1-*fosA11*-BL21 was expressed to extract the target protein, and absorbance photometry was applied for enzymatic parameter determination. Minimal inhibitory concentration (MIC) tests showed that W986 conferred fosfomycin resistance and was inhibited by phosphonoformate, thereby indicating the presence of a FosA protein. A novel resistance gene designated as *fosA11* was identified by whole-genome sequencing and bioinformatics analysis, and it shared 54.41%–64.23% amino acid identity with known FosA proteins. Cloning *fosA11* into *Escherichia coli* obtained a significant increase (32-fold) in the MIC with fosfomycin. Determination of the enzyme kinetics showed that FosA11 had a high catalytic effect on fosfomycin, with $K_m = 18 \pm 4$ and $K_{cat} = 56.1 \pm 3.2$. We also found that *fosA11* was located on the chromosome, but the difference in the GC content between the chromosome and *fosA11* was dubious, and thus further investigation is required. In this study, we identified and characterized a novel fosfomycin inactivation enzyme called FosA11. The origin and prevalence of the *fosA11* gene in other bacteria require further investigation.

**IMPORTANCE**  Fosfomycin is an effective antimicrobial agent against Enterobacterales strains. However, the resistance rate of fosfomycin is increasing year by year. Therefore, it is necessary to study the deep molecular mechanism of bacterial resistance to fosfomycin. We identified a novel chromosomal fosfomycin glutathione S-transferase, FosA11 from *Providencia rettgeri*, which shares a very low identity (54.41%–64.23%) with the previously known FosA and exhibits highly efficient catalytic ability against fosfomycin. Analysis of the genetic context and origin of *fosA11* displays that the gene and its surrounding environments are widely conserved in *Providencia* and no mobile elements are discovered, implying that FosA11 may be broadly important in the natural resistance to fosfomycin of *Providencia* species.

**KEYWORDS**  *fosA11*, fosfomycin resistance, glutathione-S transferase, *Providencia rettgeri*, whole-genome sequencing

The widespread use of antibiotics in clinical settings has led to the emergence of extensively drug-resistant and multidrug-resistant (MDR) microbes, which have become severe problems (1). Some old antibiotics are being reused in clinics to combat serious infections. In particular, fosfomycin is an antibiotic that acts as a time-dependent inhibitor of cytosolic N-acetylglucosamine enolpyruvyl transferase (MurA) by catalyzing

Address correspondence to Yongke Zheng, zyk97091@163.com, or Shenghai Wu, wu_shai@163.com.

Wei Lu and Shihan Zhou contributed equally to this article. the author order was determined by their contribution to the article.

Yongke Zheng and Shenghai Wu contributed equally to this article.

The authors declare no conflict of interest.

See the funding table on p. 13.

the first committed step of peptidoglycan synthesis. This antibiotic was commonly used in clinics due to its broad spectrum activity against a wide range of Gram-negative and Gram-positive bacteria, as well as its favorable safety profiles (2, 3). Moreover, fosfomycin has good synergistic effects with other antibiotics, including fluoroquinolones, beta-lactams, and aminoglycosides, because of its unique bactericidal mechanism (4–6). In hospitals, fosfomycin is routinely administered to treat uncomplicated urinary tract infections caused by *Enterococcus faecalis*, *Escherichia coli*, and other organisms (7–9). Due to the emergence of clinical bacteria with extensive resistance to various antibiotics, the parenteral use of fosfomycin has been investigated as a therapy for a variety of infections because it is active against many MDR pathogens (10). Fosfomycin has long been considered robust against MDR Enterobacteriaceae isolates, but few studies have investigated fosfomycin resistance in uncommon microbes. Therefore, there is an urgent need to understand the mechanisms that determine fosfomycin resistance to guide the reasonable prescription of antibiotics in practice.

Several fosfomycin resistance mechanisms have been characterized in bacteria. The most common resistance mechanisms that are relevant to clinical settings include inherent resistance to fosfomycin and chromosomal mutations that impair its transport (11). However, the most frequently observed mechanism that induces resistance involves the presence of chromosomes or plasmids that harbor genes encoding fosfomycin-modifying enzymes, which lead to bacterial resistance to fosfomycin. These enzymes can effectively inactivate fosfomycin by altering its structure. The enzymes FomB and FomA can inactivate fosfomycin by phosphorylation (12), whereas FosA, FosB, and FosX can add glutathione, l-cysteine, and $H_2O$, respectively, to its epoxide ring to cause inactivation (13).

FosA is a Mn (II)- and K+-dependent glutathione S-transferase that catalyzes the formation of a covalent bond between the sulfhydryl residue of cysteine in glutathione and C-1 in fosfomycin (14, 15). The presence of FosA and its role in fosfomycin resistance in the host strain can also be detected using the drug-sensitive disk method, which contains sodium phosphonoformate (16). Previous studies have identified *fosA* in *Serratia marcescens* (17), *fosA2* in *Enterobacter cloacae*, and *fosA3* to *fosA7* in *Escherichia coli* (16, 18–21). These genes have been identified in Gram-negative bacteria worldwide, but especially in those that cause urinary tract infections. However, few studies have aimed to identify and understand other clinically uncommon genera that simultaneously cause infection and fosfomycin resistance.

The genus *Providencia*, a phylogenetically related member of the family Enterobacteriaceae, is a Gram-negative opportunistic pathogen with eight recognized species, including *Providencia rettgeri*, *Providencia alcalifaciens*, *Providencia rustigianii*, *Providencia heimbachae*, and *Providencia stuartii* (22). In particular, *P. stuartii and P. rettgeri* are commonly isolated in urinary tract infections and known to colonize indwelling urinary catheters in residents, and they can even cause hospital outbreaks (23–25). However, according to a review by Woods and Watanakunakorn, rare cases of bacteremia caused by *Providencia* have been reported (26). Previous studies have indicated that most isolates of *Providencia* are resistant to several antibiotic agents, including penicillin, tetracycline, and polymyxin, and they probably exhibit intrinsic resistance to nitrofurantoin (27). In natural environments, *Providencia* can also exhibit resistance to β-lactams or even carbapenems by acquiring KPC and NDM genes (28–30). In general, *Providencia* exhibits customary resistance to tetracycline, older penicillins, and first- and second-generation cephalosporins, but susceptibility to aztreonam, imipenem, meropenem, and late-generation cephalosporins. Moreover, previous studies have shown that *Providencia* species vary in terms of their susceptibility to fluoroquinolones, aminoglycosides, trimethoprim-sulfamethoxazole, and fosfomycin (31, 32).

Fosfomycin resistance has been comprehensively investigated in *Klebsiella pneumoniae* and *E. coli*, but the mechanisms and prevalence of fosfomycin resistance are still unclear in other Gram-negative bacteria, especially in *Providencia* spp. Understanding the mechanisms associated with novel elements that contribute to resistance is

crucial in clinical settings. Thus, in the present study, we identified and characterized a novel fosfomycin resistance determinant called FosA11 encoded by the chromosomal glutathione S-transferase gene *fosA11* in *Providencia rettgeri* isolated from the urine of a patient.

## MATERIALS AND METHODS

### Bacterial strains, plasmids, and growth conditions

*Providencia rettgeri* strain W986 was isolated in March 2019 from the urine of an inpatient aged 80 years with a urinary tract infection at the FirHead2st Hospital of Ningbo University in Ningbo, China. Bacterial identification was preliminarily conducted with the Vitek-60 microorganism auto-analysis system (bioMérieux, Craponne, France). Further identification was conducted by comparing the average nucleotide identity with similar genera and 16S rRNA sequencing. *E. coli* BL21(DE3) was used as the host strain for expressing and purifying the FosA11 enzyme. *E. coli* DH5α was used as the recipient bacteria for routine cloning. The plasmid pUCP20 or pUCP24 was used as the vector for cloning and identifying candidate resistance genes. The pCold I vector was used for cold shock-induced expression and purifying the histidine-tagged FosA11. All of the bacterial strains mentioned above were cultured overnight at 37°C in Luria–Bertani broth containing the corresponding antimicrobial agents and/or solidified with 1.5% agar as necessary, unless stated otherwise.

### Genome sequencing, assembly, annotation, and bioinformatics analysis

Genomic DNA was extracted from *P. rettgeri* W986 using a Qiagen DNeasy PowerLyzer Microbial Kit (Qiagen Biosciences, Union City, CA, USA). Whole-genome sequencing was conducted using the Illumina HiSeq-2500 and PacBio RS II platforms by Shanghai Personal Biotechnology Co. Ltd (Shanghai, China). PacBio long reads measuring ~10–20 kb were assembled *de novo* using Canu v1.8 (33). The Illumina sequence reads were then mapped onto the primary assembly to correct possibly misidentified bases by using the Burrows–Wheeler Alignment tool (34) and Genome Analysis Toolkit (35). Annotation of the newly sequenced genome was performed by using the NCBI Prokaryotic Genome Annotation Pipeline (36), and further annotated with DIAMOND (37) against the UniProtKB/Swiss-Prot (http://web.expasy.org/docs/swiss-prot_guideline.html) and NCBI non-redundant (nr) protein databases. Annotation of resistance genes was performed using ResFinder (38) and Resistance Gene Identifier software from the Comprehensive Antibiotic Resistance Database (CARD, https://card.mcmaster.ca/). Known antimicrobial resistance genes were defined as open reading frames (ORFs) with identity >95% and coverage >85% compared with their homologs in the database (39). Multiple sequence alignment and construction of an unrooted maximum likelihood phylogenetic tree were performed with MEGA X (40) based on amino acid sequences. Phylogenetic trees were visualized alongside bar graphs of allele frequencies using the interactive web platform iTOL (41). Conserved motif analysis was conducted for the sequence of FosA11 using MEME Suite (http://meme-suite.org/). Genome visualization was performed using Gview software (42), and gene organization diagrams were drawn with Easyfig version 2.2.2 (43).

### Antibiotic susceptibility testing

The minimal inhibitory concentrations (MICs) of antimicrobial agents were determined by using the agar dilution method with Mueller–Hinton agar plates (44) according to the guidelines of the Clinical and Laboratory Standards Institute (CLSI). Susceptibility patterns were interpreted according to the CLSI breakpoint criteria and the guidelines of the European Committee on Antimicrobial Susceptibility Testing for Enterobacteriaceae. *E. coli* ATCC 25922 was included as a reference strain and quality control standard in each test run. The entire experiment was repeated three times, and all antibiotics were

human drugs obtained from the First Hospital of Ningbo University. Experimental data were analyzed by using GraphPad Prism to determine statistically significant differences.

## Cloning of resistance genes

Genomic DNA was extracted from *P. rettgeri* as described above. PCR amplification was conducted to generate DNA fragments carrying the putative resistance gene and its promoter region. The primers used are listed in Table S1 in the supplemental material. A point mutation was generated in *fosA11* by overlap extension PCR to construct the *fosA* alleles. Clonal vectors and PCR products were digested with their corresponding restriction endonucleases and ligated into pUCP20 for cloning *fosA11*, *fosA*(H7A), *fosA*(H67A), *fosA*(E113A), *aac(2ʹ)-Ia*, *aph(3ʹ)-IX*, and *ant (9)-Ia*; pUCP24 for cloning *bla-KPC-15*, *RSA-2*, *CblA-1*, *bla-OXA335*, and *bla-OXA59*; and pCold 1 for cloning the ORF of *fosA11*. The ligated products were introduced into *E. coli* DH5α or BL21(DE3) by heat shock transformation. The recombinant strains were selected on Luria–Bertani agar plates supplemented with 20 µg/mL gentamicin (for cloning genes with pUCP24 as the vector) or 100 µg/mL ampicillin (for pUCP20). The insert fragments were verified by restriction enzyme digestion and DNA sequencing.

## Expression and purification of FosA

The FosA11 protein was overexpressed by *E. coli* BL21(DE3)/pCold I::*fosA11* and purified as described previously, with minor modifications (45). Briefly, the *fosA11* gene was cloned with an His$_6$ tag on the N-terminal and enterokinase cleavage site into the pCold I vector under the control of the *cspA* promoter using the cold-shock system (46). The recombinant strain was grown overnight at 37℃ in 2 mL lysogenic broth containing 100 mg/L of ampicillin. This seed culture was then used to inoculate 200 mL of Power Prime broth (AthenaES, Baltimore, MD, USA), before incubating at 37℃. When the culture reached an optical density at 600 nm of 0.6–0.8, isopropyl-β-D-thiogalacto-side was added to a final concentration of 10 millimolar to induce protein expression, and incubation was continued for an additional 16–20 h at 15℃. About 3 g of cells was collected by centrifugation (8,000 × *g*, 10 min) at 4℃, before resuspending in 20 mM phosphate buffer at pH 7.4 supplemented with DNase (final concentration of 0.15 mU/L) in the presence of 1 mM MgSO$_4$ and disrupting by sonication. Non-disrupted cell debris was pelleted and removed by centrifugation at 8,000 rpm for 30 min. The lysates were blended with pre-equilibrated nickel–nitrilotriacetic acid agarose resin (Beyotime Biotechnology, Shanghai, China) and kept overnight at 4℃ with gentle shaking. The mixture containing the FosA11 protein was then loaded onto a column and purified according to the instructions provided with a His-tag Protein Purification Kit (P2226, Beyotime, China). The His$_6$ tag was removed by incubating overnight with enterokinase at 15℃. The digested FosA11 was purified using a 3 kDa ultrafiltration tube to retain the target protein and remove the free His$_6$ tag. The purity of FosA11 was checked by sodium dodecyl sulfate-polyacrylamide gel electrophoresis, and the protein concentration was determined using a BCA protein assay kit (Thermo Fisher Scientific, Rockford, IL, USA) and spectrophotometer.

## Enzyme kinetics

A kinetic assay was conducted to monitor the activity of FosA by fosfomycin-dependent glutathione conjugation. The reaction was monitored spectrophotometrically using monochlorobimane (Sigma-Aldrich, St. Louis, MO, USA), as reported previously with slight optimizations (47). Briefly, experiments were conducted in a volume of 50 µL at 25℃ for 20 min (for various concentrations of glutathione) or 30 min (for various concentrations of fosfomycin) in 0.1 M HEPES buffer at pH 8.0 containing 0.05 mM MnCl$_2$, various concentrations of glutathione (0–60 mM), fosfomycin (0–20 mM), and 250 nM FosA6. A blank control without enzyme was also performed. Reactions were ended by adding 100 µL methanol. Quenched reactions were diluted 10 times in 100 µL of 0.1 M sodium phosphate buffer pH 8.0 containing 1 mM EDTA. After adding 500 µM

monochlorobimane, the concentration of glutathione was determined by fluorescence spectroscopy using a SpectraMax M2 Plate reader (Molecular Devices). A standard curve was produced by adding 0–750 µM glutathione. Data were fitted to the Michaelis–Menten equation using SigmaPlot (Systat Software Inc, San Jose, CA, USA).

## RESULTS AND DISCUSSION

### Antibiotic susceptibility of *P. rettgeri* isolate W986

*Providencia rettgeri* W986 was isolated from an inpatient with a urinary tract infection in September 2020. This patient exhibited a weak anti-infection effect after treatment with fosfomycin combined with cefuroxime and was then cured with a combination of imipenem and nitrofurantoin. Initially, this strain was noted by the clinical laboratory as sensitive to meropenem, imipenem, trimethoprim-sulfamethoxazole, ciprofloxacin, and amikacin but resistant to colistin, first- and second-generation cephalosporins, and fosfomycin.

Similar to previous uncommonly reported infections caused by *P. rettgeri* (48), MDR *P. rettgeri* isolates have rarely been encountered in hospitals, and thus the antibiotic susceptibility of W986 was also assessed using the agar dilution method. The antibiotic resistance phenotype of *P. rettgeri* W986 is shown in Table S2. In general, the MICs for this strain indicated strong resistance to polymyxin, ampicillin, fosfomycin, and streptomycin, but sensitivity to ceftazidime, trimethoprim, imipenem, and tigecycline. Furthermore, the resistance of *P. rettgeri* W986 to fosfomycin was confirmed by a disk potentiation test under saturation with phosphonoformate, a specific inhibitor of FosA enzyme, as described by Nakamura et al. (16). Figure 1 shows that W986 possessed a fosfomycin-inactivating enzyme with a critical role in protecting the host strain against fosfomycin, and it was evidently inhibited by phosphonoformate (Fig. 1). In summary, these results agree with those obtained in the aforementioned clinical trials and further demonstrated the MDR phenotype of strain W986. Previous studies have shown that most clinical strains of *P. rettgeri* exhibit extensive resistance to β-lactams (29, 49) and polymyxin B and E (50), but susceptibility to aminoglycosides (29) and ciprofloxacin (31). These findings

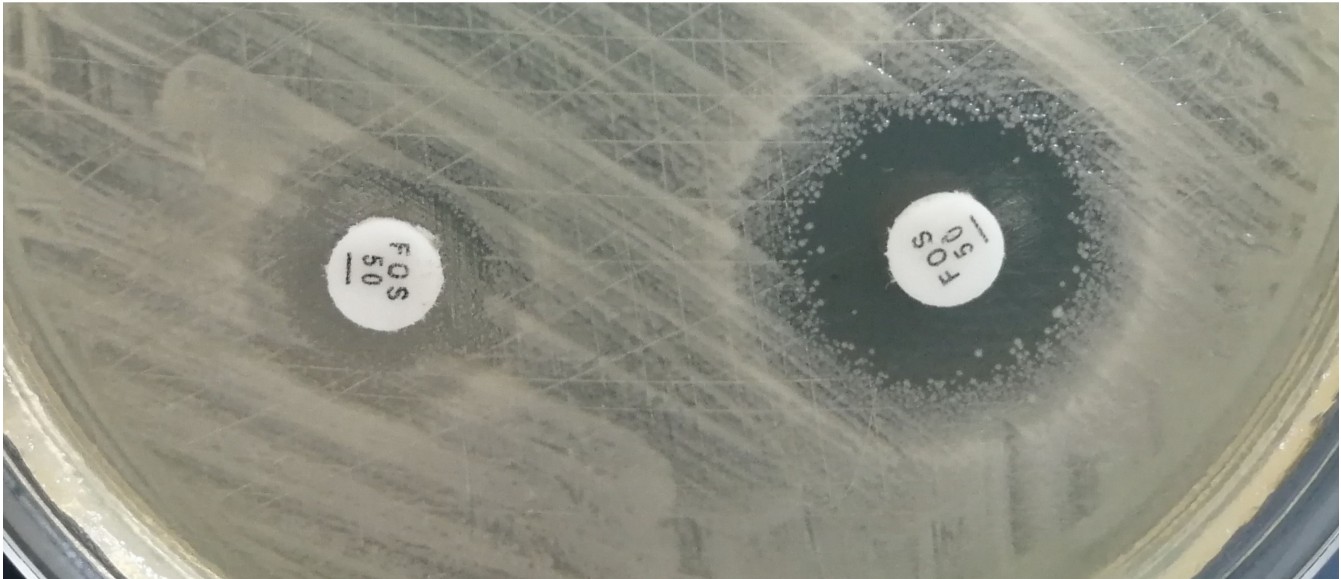

**FIG 1** Kirby-Bauer drug susceptibility test. Inhibition zone sizes by fosfomycin (50 µg/disk) on MH-G6P agar plates in the Kirby-Bauer drug susceptibility test. The *Providencia rettgeri* W986 harboring *fosA11* on Mueller-Hinton agar plate; the fosfomycin disk on the left contained 50 g of glucose-6-phosphate (G6P), and the disk on the right contained 50 g of G6P in addition to 1 mg of phosphonoformate.

regarding the MDR phenotype of W986 are consistent with previous reports, but it also exhibited a broader resistance phenotype.

## Genome characteristics of strain W986

To obtain insights into the molecular basis of the MDR phenotype of W986, the whole genome of this strain was sequenced. The genome of W986 comprised 4.5 Mbp with only one circular chromosome (GenBank accession number: CP076258) and an average GC content of 40.23%. Annotation of the W986 chromosome identified 3,975 protein-coding sequences with 22 rRNA and 77 tRNA genes (Table 1 and Fig. 2). In the publicly available genome sequences of *Providencia* species, the highest whole-genome average nucleotide identity value (99.12%) was found with *P. rettgeri* FDAARGOS_330 (NCBI assembly database accession number: GCA_002984195), thereby indicating that W986 is most closely related to this strain, which is consistent with the results obtained by

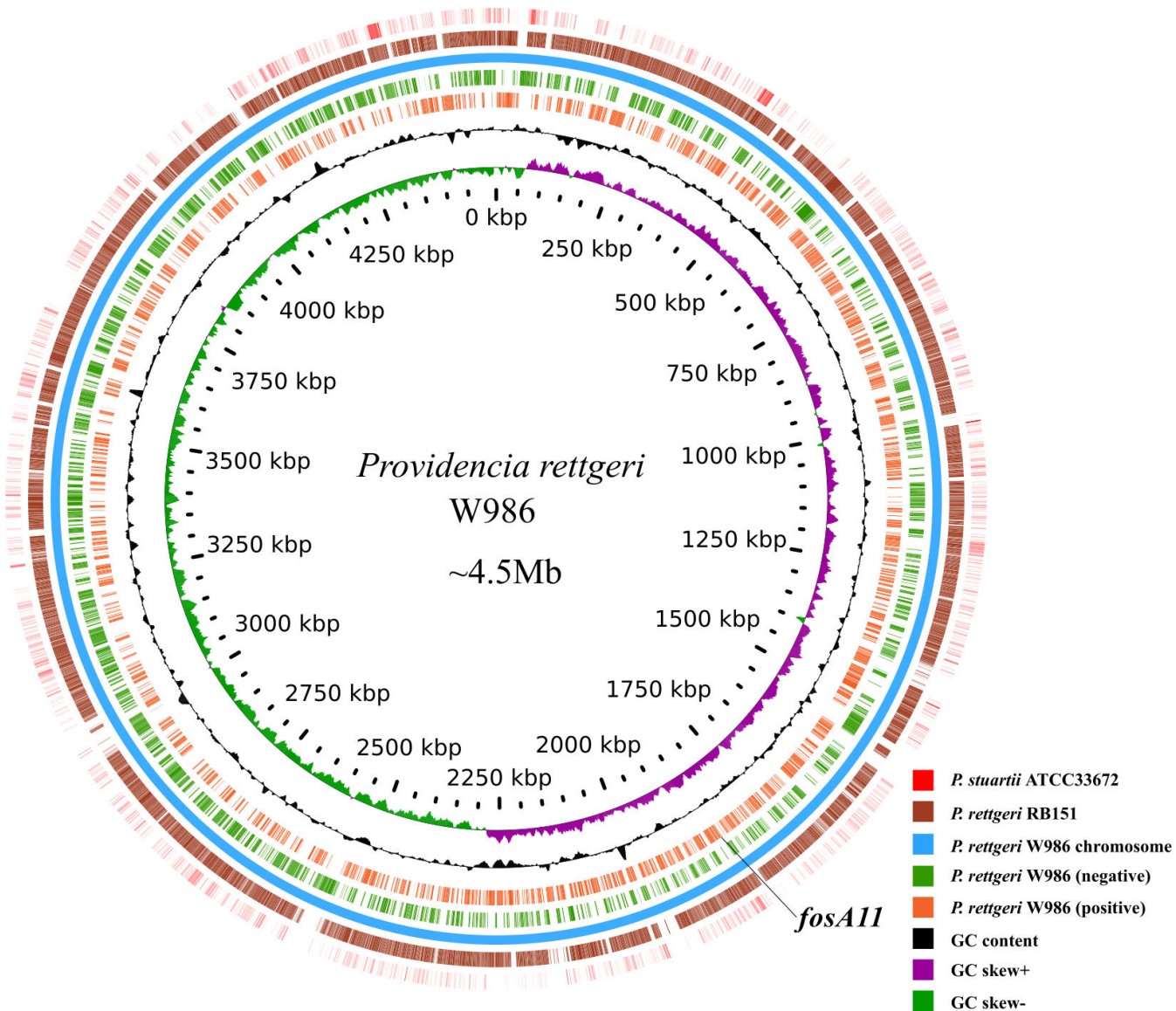

**FIG 2** Genomics feature of *Providencia rettgeri* W986. Circular representation of chromosome of *P. rettgeri* W986 (GenBank accession number: CP062758). The black arrow indicates the chromosomal location of *fosA11*. From outside to inside: circle 1, representation of the *Providencia stuartii* ATCC33672 genome; circle 2, circular representation of the *P. rettgeri* RB151 genome; circle 3, circular representation of the *P. rettgeri* W986 genome; circles 4 and 5, predicted ORFs encoded in the plus strand and minus strand, respectively; circles 6 and 7, GC content and GC skew maps, respectively; and circle 8, scale in kb (each tick is 50 kb).

**TABLE 1** General features of the genome of *Providencia rettgeri* W986

| Characteristics | Chromosome |
|---|---|
| Size (bp) | 4,515,465 |
| G + C% | 40.23 |
| Genes | 4,147 |
| ORFs | 4,044 |
| Known proteins | 3,975 |
| Hypothetical proteins | 69 |
| Average ORF length (bp) | 1,029 |
| Average protein length (aa) | 343 |
| ncRNAs | 4 |
| tRNAs | 77 |
| rRNAs | 16S: 7; 23S: 7; and 5S: 8 |

genotyping based on 16S rRNA sequencing and housekeeping gene analysis. Comparative genomics analysis showed that the genome of *P. rettgeri* W986 shared the highest sequence similarity with that of *P. rettgeri* RB151 (accession number: CP017671; 99.06% identity and 96.3% coverage) but low sequence identity with *P. stuartii* ATCC33672 (accession number: CP008920; 76.56% identity and 63.1% coverage) (Fig. 2)

To identify known resistance genes, gene variants, and putative novel resistance genes related to the bacterial resistance phenotype, the predicted ORFs were BLASTed against CARD and the NCBI nr protein database. Remarkably, no known antibiotic resistance genes related to the bacterial resistance phenotype could be inferred from the coding sequences. Therefore, nine unique ORFs were identified with variable similarities to the closest known resistance genes (Table 2). All of these candidate novel genes were then selected for further functional verification.

## Functional verification of candidate novel resistance genes and characterization of novel fosfomycin glutathione-transferase gene *fosA11*

Plasmid-borne *fosA* variants have been described in *Klebsiella pneumoniae*, *E. coli*, and *Serratia* spp. (17). However, little is known about the characteristics and prevalence of *fosA* in other Enterobacteriaceae species. Thus, we performed whole-genome sequencing for fosfomycin-resistant strain W986 to characterize its resistance mechanism. However, no plasmids related to multi-drug resistance were found, and the annotation results indicated no related known drug resistance genes, so we considered that *P. rettgeri* W986 may have evolved novel drug resistance determinants. Therefore, we selected nine candidate resistance genes that could be related to antibiotic resistance according to our results (Table 2) and cloned them into a high-copy number vector, before transformation into *E. coli* DH5α (Table S3). Antimicrobial susceptibility tests were conducted to identify the potential resistance genes. Intriguingly, except for *fosA11*, expressing all of the other candidate genes did not lead to increases in the MIC values against the corresponding drugs. Remarkably, the MIC value for the clonal strain of *fosA11* increased by 128-fold compared with that for the control strain, and the difference between the clone and control strains was highly significant ($P$-value < 0.01, Student's *t*-test). Finally, we identified a novel fosfomycin resistance gene designated as *fosA11* (Table S3; Table 3). Furthermore, the MIC test results indicated that the antibiotic sensitivity of *Providencia* did not seem to be affected by glucose-6-phosphate (G6P). Drug susceptibility testing with other *Providencia* strains also showed that none were affected by the addition of G6P (data not shown). Based on these results, we searched for several genes in the Uhp system using the NCBI database to perform BLAST analysis against our sequenced W986 strain and found no related genes. Thus, more research is required to clarify the molecular mechanism involved.

The *fosA11* gene had a length of 414 bp and it encoded a protein comprising 138 amino acids. The highest amino acid sequence identity (64.23%) was shared with the plasmid-encoded fosfomycin glutathione transferase FosA5, which was detected

**TABLE 2** Putative novel resistance genes and antibiotic resistance phenotypes

| Antibiotics resistance pattern | Putative novel resistance genes | Length (bp) | Identity (%) | Observed resistance phenotypes[b] | GenPept accession no. of closest homolog |
|---|---|---|---|---|---|
| Aminoglycosides | *aac(2′)*-like | 537 | 65.17 | –[a] | WP_174822168 [aac(2′)-Ia] |
|  | *aph(3′)*-like | 774 | 52.94 | – | BBT16429 [aph(3′)-IX] |
|  | *Spd*-like | 678 | 54.55 | – | WP_021285909 (spd) |
| Fosfomycin | *fosA*-like (*fosA11*) | 414 | 64.23 | Fosfomycin | KP143090 (fosA5) |
| β-lactams | *bla*$_{RSA}$-like | 660 | 70.59 | – | MG739504 (RSA-2) |
|  | *bla*$_{KPC}$-like | 993 | 60.87 | – | AAK70220 (KPC-15) |
|  | *bla*$_{SRT}$-like | 1,143 | 55.99 | – | ABQ52459 (SRT-2) |
|  | *CblA*-like | 945 | 57.14 | – | AMP48925 (CblA-1) |
|  | *bla*$_{OXA}$-like | 828 | 52.94 | – | AGW16417 (OXA-335) |

[a]–, no resistance to specific antibiotics observed when expressed in *E. coli* DH5α.
[b]Antibiotic resistance profiles of recombinant *E. coli* DH5α with each putative resistance gene identified in this study.

in an extended-spectrum β-lactamase (ESBL)-producing *E. coli* isolate from Shanghai, China (19). The second closest relatives of *fosA11* comprised *fosA10*, *fosA9*, and *fosA3*, with 63.5% shared amino acid sequence identity. Thus, the resistance determinant was designated as *fosA11* to maintain consistency with the nomenclature of the latest *fosA10* gene described recently by Huang et al. (51). *fosA10* is the 11th member of the *fosA* family to be identified, and it was the focus of the rest of this study. BLASTp searches against the NCBI database detected the protein (100.00% shared identity) in multiple *Providencia* species (GenPept accession no.: WP_042844077.1), and it shared 81.75% and 82.48% amino acid identity with the putative fosfomycin glutathione transferases encoded in the chromosomes of *Providencia huaxiensis* (accession no.: WP_206275859.1) and Enterobacterales (accession no.: WP_004263671.1), respectively. The results also showed that among 96 sequenced genomes, 61 encoded *fosA11*. When expressed in *E. coli* DH5α, *fosA11* conferred resistance to fosfomycin with an MIC of 128 µg/mL. The MIC values increased 16- and 32-fold compared with the control strains DH5α and DH5α carrying the vector pUCP20, respectively, thereby suggesting that *fosA11* had the predicted function (Table 3). However, interestingly, when expressed in BL21, this gene led to more obvious resistance to fosfomycin, where the MIV values increased 128-fold on agar containing G6P and 32-fold without G6P compared with the control strain (BL21 carrying the vector pCold1) (Table 3). This difference may have been due to strong regulation of the protein expression system in BL21 and the production of a large amount of the target protein under intense expression of the target gene, thereby leading to more efficient inactivation of fosfomycin, and thus the MIC value increased greatly compared with the recombinant DH5α.

In addition, a phylogenetic tree constructed containing all previously identified fosfomycin glutathione transferases collected from the NCBI nr and CARD databases showed that this protein clustered closest to FosA-type enzymes (Fig. 3).

**TABLE 3** Antimicrobial susceptibility test results (MICs) for fosfomycin against *Providencia rettgeri* W986 and clonal strain[a]

| Strains | Characteristics | Fosfomycin with G6P (mg/L) | Fosfomycin without G6P (mg/L) |
|---|---|---|---|
| ATCC25922 | Quality control strain | 2 | 8 |
| *Providencia rettgeri* W986 | Contains original *fosA11* | 1,024 | 1,024 |
| DH5α | Cloned recipient strain | 2 | 16 |
| DH5α/pUCP20 | Vector control | 2 | 16 |
| DH5α/pUCP20/*fosA11* | Vector harboring *fosA11* | 64 | 128 |
| Fold change |  | 32 | 16 |
| BL21(DE3) | Cloned recipient strain | 2 | 16 |
| BL21/pCold1 | Expression vector control | 2 | 16 |
| BL21/pCold1/*fosA11* | Vector harboring *fosA11* | 256 | 512 |
| Fold change |  | 128 | 64 |

[a]MICs were determined by using the agar dilution method with or without 25 mg/L glucose-6-phosphate.

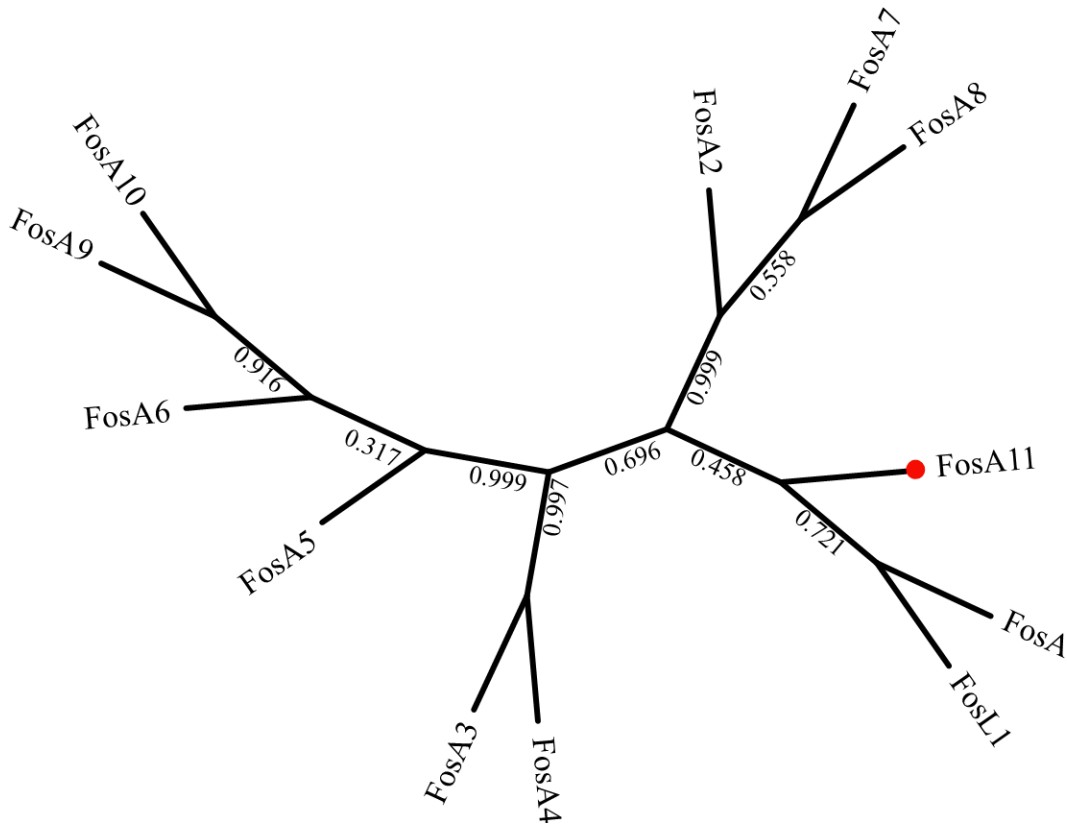

**FIG 3** FosA phylogenetic tree. Phylogenetic analysis of FosA11 and all other 11 known FosA proteins. GenBank accession numbers are listed in Fig. 2. The bootstrap values are shown at the nodes of the tree. The scale bar represents a 10% amino acid sequence difference. FosA11 from this study is highlighted with a red-filled circle.

Thus, the FosA11 protein had a close relationship with FosA and FosL1 in *Salmonella enterica* and *E. coli*, respectively (52), and multiple sequence alignment showed that FosA11 shared 54.41%–64.23% amino acid sequence identity with previously reported FosA subtypes, and it contained several conserved residues and characteristic sequence motifs found in FosA enzymes (Fig. 4).

## Functional characterization and kinetic parameters of FosA11 glutathione-S-transferase enzyme

To analyze the levels of resistance mediated by different FosA enzymes, we cloned the *fosA3* gene into pUCP20-transformed *E. coli* DH5α as a positive control for comparison with *fosA11*. The MICs obtained for the *fosA3* transformants were >1,024 µg/mL, but that for the *fosA11* transformant was only 64 µg/mL. We then determined the steady-state kinetic parameters for FosA3 and FosA11 using recombinant purified enzymes to separate the contributions to the resistance of FosA expression and activity. The kinetics data showed that FosA11 could inactivate fosfomycin with an affinity ($K_M$ value) of 18 ± 4 mM and catalytic efficiency ($K_{cat}/K_M$) of 2.9 ± 0.5 × 10$^3$ M$^{-1}$ s$^{-1}$. The data also indicated that FosA11 was less active than FosA3. The increases in the activities of FosA3 and

**TABLE 4** Kinetic parameters for FosA11 and FosA3 in assessments of fosfomycin[a] inactivation

| Enzyme | MIC (µg/mL) | $k_{cat}$ (s$^{-1}$) | $K_m$ (mM) | $k_{cat}/K_m$ (M$^{-1}$ s$^{-1}$) |
|---|---|---|---|---|
| FosA3 | 1,024 | 112.4 ± 5.3 | 12 ± 2 | (9.0 ± 1.0) × 10$^3$ |
| FosA11 | 256 | 56.1 ± 3.2 | 18 ± 4 | (2.9 ± 0.5) × 10$^3$ |

[a]The steady-state kinetic parameters for FosA11 and FosA3 with fosfomycin were determined in the presence of 30 mM glutathione. Kinetic parameters are reported as means ± standard deviations based on three or four independent biological replicates.

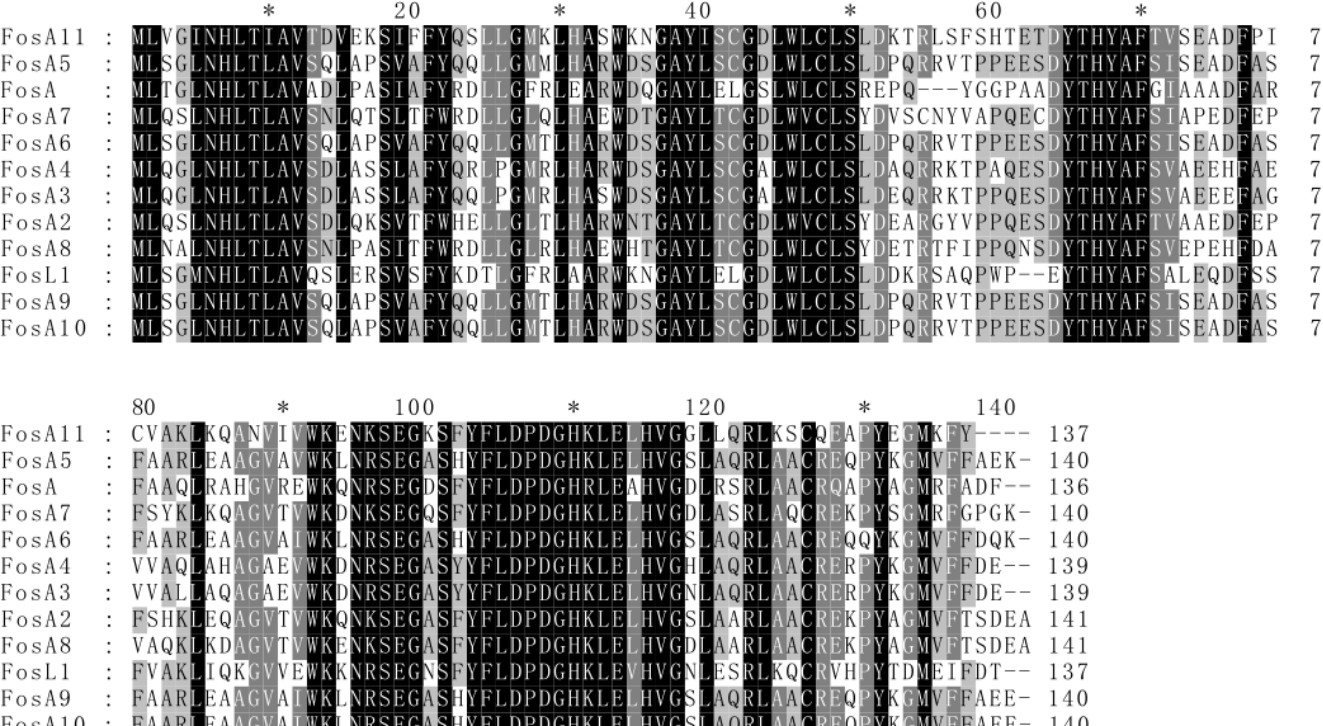

**FIG 4** FosA11 sequences and their comparison to previously identified FosA. Multiple sequence alignment of FosA proteins. Amino acid sequence alignment of FosA11 (MZ277617), FosA (AAA98399), FosA2 (ACC85616;), FosA3 (AB522970), FosA4 (AB908992), FosA5 (AJE60855), FosA6 (NG051497), FosA7 (KKE03230), FosA8 (CP013990), FosA9 (PRJEB32329), FosA10 (MT074415), and FosL1 (QHR93773). The alignment was obtained using MEGA X. The black background indicates fully conserved residues; gray background indicates strongly similar residues. The numbers on the right correspond to the amino acid residues in each full-length protein.

FosA11 were mainly driven by an approximately twofold increase in $K_{cat}$, with a slight difference in $K_M$ for fosfomycin (Table 4). Overall, these data demonstrate that FosA3 exhibited greater fosfomycin resistance than FosA11 under the expression of the protein in *E. coli*, and the enzymatic activity of drug-resistance enzyme FosA11 was at a moderate catalytic level compared with FosA3. Hence, we suggest that this enzyme may also have a catalytic function related to glutathione transfer in *P. rettgeri* W986. Therefore, as mentioned above, we cannot categorically state that FosA11 is a drug-resistance protein with a primary catalytic function in fosfomycin resistance, and further study is required.

## Amino acid residues critical for function of FosA

The FosA enzyme has a single mononuclear $Mn^{2+}$ site in each subunit, and it catalyzes the addition of glutathione to the oxirane ring of the antibiotic to render it inactive. Previous structural and mutagenesis studies have demonstrated the importance of several residues that are absolutely conserved in FosA proteins at the primary and tertiary structure levels, as well as the functional levels (53). In particular, three equivalent amino acid residues in FosA11 comprising His7 (H7), His67 (H67), and Glu113 (E113) were predicted by InterPro and UniProt Knowledgebase as critical for the appropriate function of this enzyme (53).

To investigate the specific roles of the three residues comprising H7, H67, and E113 in the function of FosA11, we performed site-directed mutagenesis to individually replace them with alanine residues and examined their effects on the levels of resistance to fosfomycin. Compared with the wild-type FosA11, all of the substitutions completely abolished the resistance to fosfomycin in the mutant strains (Table 5). These results indicate that the residues H7, H197, and E210 are critical for the function of FosA11. Furthermore, these findings agree well with previous suggestions of the significant

**TABLE 5** MIC values obtained in assessments of effects of site-directed mutagenesis on drug resistance function of fosA11[a]

| Strains | Characteristics | Fosfomycin with G6P (mg/L) | Fosfomycin without G6P (mg/L) |
|---|---|---|---|
| ATCC25922 | Quality control strain | 2 | 8 |
| DH5α | Cloned recipient strain | 2 | 16 |
| DH5α/pUCP20 | Vector control | 2 | 8 |
| DH5α/pUCP20/*fosA11* | Wildtype of *fosA11* | 64 | 128 |
| DH5α/pUCP20/*fosA11*(H7A) | Mutagenesis *fosA11* | 2 | 4 |
| DH5α/pUCP20/*fosA11*(H67A) | Mutagenesis *fosA11* | 4 | 4 |
| DH5α/pUCP20/*fosA11*(E113A) | Mutagenesis *fosA11* | 2 | 4 |

[a]MICs were determined by using the agar dilution method with or without 25 mg/L glucose-6-phosphate.

contributions of the equivalent residues to recognition, catalysis, and resistance in other FosA proteins, such as FosA[PA] (54) and FosA3 (55). However, to obtain a clearer understanding of the specific roles of these FosA11 residues in catalysis and substrate recognition, additional studies will be required by combining mutagenesis, biochemical analyses, and X-ray crystallography to characterize the structural and molecular basis that underlies the modification of aminoglycoside substrates by the FosA enzyme, which is of critical importance for the accurate design of inhibitors.

## Analysis of genetic environment and origin of *fosA11*

The *fosA11* gene was located on the chromosome in *P. rettgeri* W986 (Fig. 5) according to whole-genome sequencing. To understand the genetic context of *fosA11*, both sides of the flanking regions around *fosA11* at approximately 40 kb and closely related *fosA*-encoding gene regions in five other *Providencia* strains were analyzed, as shown in Fig. 5. No mobile genetic elements were found in the neighboring regions of *fosA*. Moreover, the GC content of *fosA11* (59.43%) was inconsistent with that of the chromosome (40.23%) in W986. Comparative genomic analysis of the flanking region of *fosA11* in W986 and *fosA11*-carrying fragments in other *Providencia* species showed that all *fosA11* genes were surrounded by two genes encoding hypothetical proteins. The adjacent regions of *fosA* in W986 (especially the *mntB_1* at the 5′ end of *fosA11* to *infC_2* at the 3′ end of *fosA11*) shared significant sequence similarity with the corresponding parts of the chromosome sequences in these *Providencia* strains (Fig. 5), although the *pheS-arnT* fragment at the 3′ end of *fosA11* was not present in the corresponding chromosome region in *P. huaxiensis* WCHPr (Fig. 5). These results indicate that *fosA11* may be an intrinsic feature in *Providencia*. However, the primary function of FosA11 in *Providencia* still needs to be determined, but we suggest that the ability to modify fosfomycin might be one of its normal functions and vital for protecting against naturally occurring fosfomycin in the environment for millennia before the antibiotic era began.

Previous studies have comprehensively demonstrated that the *fosA* gene can be located on plasmids or chromosomes, but it should be noted that the dissemination of *fosA* has caused clinical therapy problems, e.g., the most frequently reported plasmid-mediated *fosA3* is widely distributed among ESBL-producing *E. coli* in East Asia (17). Our findings suggest that *fosA11* is representative of a vast reservoir of fosfomycin resistance determinants that may spread to non-FosA-producing species such as *E. coli* because fosfomycin is used extensively in the clinic.

## Conclusion

In this study, we identified a novel fosfomycin resistance gene in the chromosome of a *P. rettgeri* W986 isolate from a hospital and demonstrated that it encodes a functional enzyme called FosA11, which can effectively inactivate fosfomycin. No known mobile genetic elements were found in the genetic environment of *fosA11*, but FosA analogs may comprise an enormous reservoir of fosfomycin resistance elements that could be transferred to non-FosA-producing organisms such as *E. coli* in the era of antibiotics abuse. These findings also indicate that inhibiting the activity of FosA may be a feasible

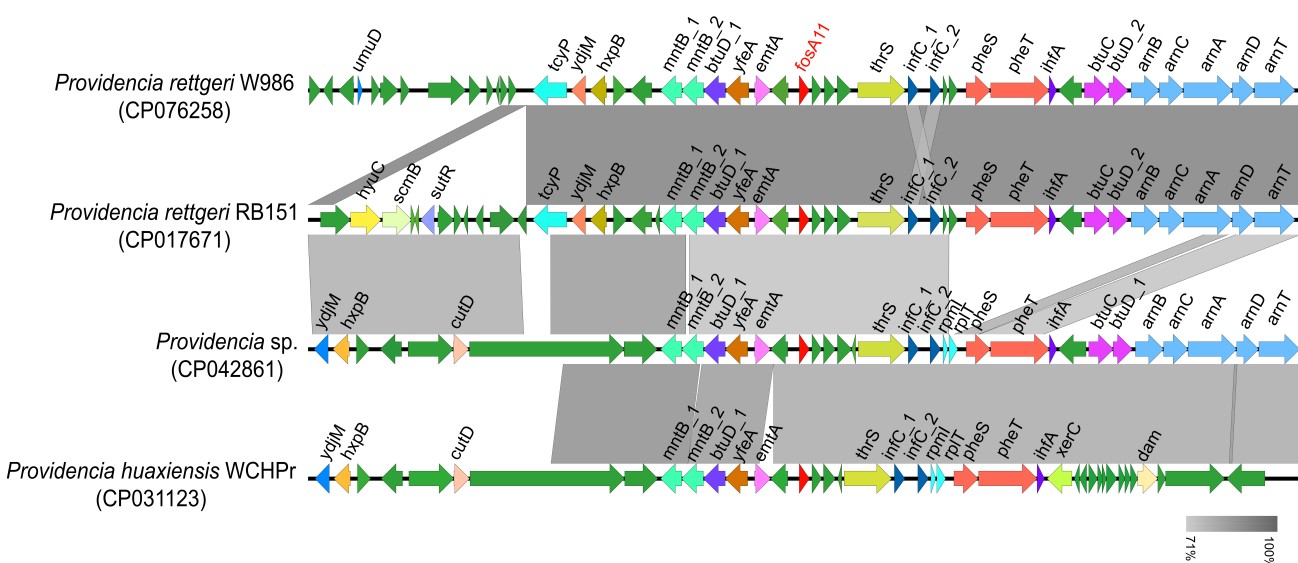

**FIG 5** Genetic environment of fosA11. Schematic representation of the genetic environment of *fosA11* and comparison of the *fosA11*-carrying regions in genomes of *Providencia* strains. ORFs are shown as arrows drawn to scale to indicate the direction of transcription. The *fosA11* gene is colored in red, and the other genes with known function are colored in corresponding color. The ORFs encoding hypothetical proteins are colored in green. *umuD*, protein UmuD; *tcyP*, L-cystine uptake protein; *ydjM*, membrane protein; *hxpB*, hexitol phosphatase B; *mntB_1*, manganese transport system membrane protein; *mntB_2*, manganese transport system membrane protein; *btuD_1*, vitamin B12 import ATP-binding protein; *yfeA*, periplasmic chelated iron-binding protein; *emtA*, endo-type membrane-bound lytic murein transglycosylase A; *fosA11*, fosfomycin glutathione-S-transferase FosA11; *thrS*, threonine–tRNA ligase; *infC_1*, translation initiation factor IF-3; *infC_2*, translation initiation factor IF-3; *rpmI*, 50S ribosomal protein L35; *rplT*, 50S ribosomal protein L20; *pheS*, phenylalanine–tRNA ligase alpha subunit; *pheT*, phenylalanine–tRNA ligase beta subunit; *ihfA*, integration host factor subunit alpha; *btuC*, vitamin B12 import system permease protein; *btuD_2*, vitamin B12 import ATP-binding protein; *arnB*, UDP-4-amino-4-deoxy-L-arabinose–oxoglutarate aminotransferase; *arnC*, undecaprenyl-phosphate 4-deoxy-4-formamido-L-arabinose transferase; arnA, bifunctional polymyxin resistance protein; *arnD*, putative 4-deoxy-4-formamido-L-arabinose-phosphoundecaprenol deformylase; *arnT*, undecaprenyl phosphate-alpha-4-amino-4-deoxy-L-arabinose arabinosyl transferase.

approach for broadening the activity of fosfomycin beyond *E. coli* to combat extensively drug-resistant bacteria such as *Klebsiella* and *Enterobacter* spp. However, FosA might not exclusively inactivate fosfomycin, and thus the primary function of FosA remains uncertain. In addition, the origin of *fosA11* and its metabolic significance require further study. We consider that our findings will contribute significantly to understanding resistance mechanisms and promote clinical infection control for uncommon fosfomycin-resistant bacteria.

## ACKNOWLEDGMENTS

The authors would like to acknowledge all study participants and individuals who contributed to this study.

This study was supported by the Science & Technology Project of Wenzhou City, China (N20210001 and Y2020112), Zhejiang Provincial Natural Science Foundation of China (LY19C060002 and LQ17H190001), and Natural Science Foundation of China (81973382).

The study was conceptualized and designed by S.W. and W.L. Data acquisition was done by W.L. and X.M. W.L. performed the experiments, analyzed and interpreted the data, and drafted the manuscript.

The authors affirm that this manuscript is an honest, accurate, and transparent account of the study being reported; that no important aspects of the study have been omitted; and that any discrepancies from the study as planned (and, if relevant, registered) have been explained. The authors declare that the research was conducted

in the absence of any commercial or financial relationships that could be construed as a potential conflict of interest.

## AUTHOR AFFILIATIONS

[1]Department of Laboratory, Affiliated Hangzhou First People's Hospital, School of Medicine, Westlake University, Hangzhou, China
[2]The Fourth School of Medicine Affiliated to Zhejiang Chinese Medical University, Hangzhou, China
[3]Department of Intensive Care Unit, Affiliated Hangzhou First People's Hospital, School of Medicine, Westlake University, Hangzhou, China

## AUTHOR ORCIDs

Wei Lu http://orcid.org/0009-0006-8953-9572
Shenghai Wu http://orcid.org/0000-0002-9923-1014

## FUNDING

| Funder | Grant(s) | Author(s) |
|---|---|---|
| Zhejiang Provincial Medical and Health Technology Project | 2024KY229 | |
| Medical and Health Technology Project of Hangzhou | ZD20220012 | |

## AUTHOR CONTRIBUTIONS

Shihan Zhou, Data curation, Investigation | Xueli Ma, Resources | Dongxin Liu, Investigation, Resources | Keqing Zhang, Software | Yongke Zheng, Investigation, Resources | Shenghai Wu, Conceptualization, Investigation, Methodology.

## DATA AVAILABILITY

The nucleotide sequence data reported in this study have been submitted to Gen-Bank under accession number CP076258 for the chromosome of *P. rettgeri* W986, and MZ277617 for *fosA11*.

## ETHICS APPROVAL

This study did not involve individual patient data, and only anonymous clinical residual samples obtained during routine hospital laboratory procedures were used. This study was approved by the ethics committee of the Hangzhou First People's Hospital, Hangzhou, Zhejiang, China.

## ADDITIONAL FILES

The following material is available online.

### Supplemental Material

**Table S1 to S3 (Spectrum02542-23-S0001.docx).** Supplemental tables.

### Open Peer Review

**PEER REVIEW HISTORY (review-history.pdf).** An accounting of the reviewer comments and feedback.

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
