## [Reviewer comments · Microbiology Spectrum]

Microbiology Spectrum

fosA11*, a Novel Chromosomal Encoded Fosfomycin Resistance Gene Identified in *Providencia rettgeri

Wei Lu, Shihan Zhou, Xueli Ma, Nuo Xu, Dongxin Liu, Keqing Zhang, Yongke Zheng, and Shenghai Wu

Corresponding Author(s): Shenghai Wu, Hangzhou First People's Hospital

Review Timeline:

Submission Date:	June 22, 2023
Editorial Decision:	August 10, 2023
Revision Received:	October 28, 2023
Accepted:	November 29, 2023

Editor: Silvia Cardona

Reviewer(s): Disclosure of reviewer identity is with reference to reviewer comments included in decision letter(s). The following individuals involved in review of your submission have agreed to reveal their identity: Maria Hadjifrangiskou (Reviewer #2)

Transaction Report:

DOI: <https://doi.org/10.1128/spectrum.02542-23>

August 10, 2023

Mr. Shenghai Wu
Hangzhou First People's Hospital
Hangzhou
China

Re: Spectrum02542-23 (*fosA11*, a Novel Chromosomal Encoded Fosfomycin Resistance Gene Identified in *Providencia rettgeri*)

Dear Mr. Shenghai Wu:

Thank you for submitting your manuscript to Microbiology Spectrum. Your manuscript was evaluated by two experts in the field who pointed out issues that need to be addressed in a revised version.

Link Not Available

Sincerely,

Silvia Cardona

Journals Department
Reviewer comments:

Reviewer #1 (Comments for the Author):

In this study, authors first identified and characterized a novel Fosfomycin inactivate enzyme, named FosA11. The function has been confirmed using a cloning experiment and Kinetic assay was conducted. The genetic structure has been presented. In fact, some studied has indicated that chromosomal FosA is widely distributed among Gram-negative pathogenic species including *Providencia stuartii* and contributes to intrinsic fosfomycin resistance (PMID: 28851843). I wonder whether this gene is inherent and whether the name *fosA11* is proper.

The minor comment is that the Table 1 could be transferred to supplement materials.

Reviewer #2 (Comments for the Author):

In this work, Lu et al provide novel insights into fosfomycin resistance in *Providencia rettgeri*. Specifically, using a UTI *P. rettgeri* isolate, *P. rettgeri* W986, the authors present work on a novel fos resistance gene, which they name *fosA11*. They provide evidence of catalytic activity against fosfomycin, although *FosA11*'s affinity for fosfomycin is lower than the characterized *FosA3* from *E. coli*. This study reports that this novel gene, *fosA11* is chromosomally-encoded and prevalent among *Providencia* species, which could explain intrinsic resistance to fosfomycin in such isolates. Overall the study is linearly performed and provides new insight into the field. However, there are areas that lack rigor, which need to be addressed.

- 1) There are no statistical analyses for the MICs and more extensive description of how experiments were performed would be useful.
- 2) Critical information shown in the figures and tables is oftentimes not discussed in the main text. For example, it is not mentioned in the text that G6P is not required for uptake of fosfomycin by *Providencia*. However this is demonstrated in Table 3. Does *Providencia* not have the Uhp system described in *E. coli*? This would be good to discuss, given that *Providencia* is understudied and the authors have performed whole genome sequencing
- 3) There is no description of how the candidate fosfomycin resistance genes were identified (Lines 297-300). As mentioned above, it would be more useful to provide a table of the genes that are putatively associated with fosfomycin or other drug resistance in the main text rather than the supplement.
- 4) There should be some genetic validation with mutagenesis of the catalytic pocket of *FosA11* to validate that this mutant stops imparting fosfomycin resistance to *E. coli*.
- 5) The authors state that there are numerous *Providencia* species that have the *fosA11*. Since the authors have run the analyses, they should be quantitative here: "out of xx sequenced genomes, Y number of genomes code for *fosA11*".

Minor

The article needs editing for English. In some cases, like in lines 238-241 it is hard to understand what the authors meant to state.

Please make sure that the references are separated by a space between the reference and the last word in a sentence.

The MICs for the other drugs for strain W986 should be given in the form of a table instead of described in the text. (lines 241-248)

In materials and methods, did you mean 10 micromolar or 10 millimolar IPTG?

Staff Comments:

Preparing Revision Guidelines

Please return the manuscript within 60 days; if you cannot complete the modification within this time period, please contact me. If you do not wish to modify the manuscript and prefer to submit it to another journal, please notify me of your decision immediately so that the manuscript may be formally withdrawn from consideration by Microbiology Spectrum.

Response to comments of the referees

Spectrum02542-23 (*fosA11*, a Novel Chromosomal Encoded Fosfomycin Resistance Gene Identified in *Providencia rettgeri*)

Editor: Silvia Cardona

>>**RESPONSE:** Thank you for your favorable evaluation of our manuscript, and we thank the reviewer for his/her comments and feedback. We have studied the comments carefully and have made revisions accordingly which we hope meet with approval. Changes are highlighted in yellow using the highlight tool in the revised manuscript to make the revisions notable, and point-by-point responses to the reviewers' comments are listed below.

Comments of the referees:

Referee: 1 (Comments to the Author):

>>**COMMENT:** In this study, authors first identified and characterized a novel Fosfomycin inactivate enzyme, named FosA11. The function has been confirmed using a cloning experiment and Kinetic assay was conducted. The genetic structure has been presented. In fact, some studies have indicated that chromosomal FosA is widely distributed among Gram negative pathogenic species including *Providencia stuartii* and contributes to intrinsic fosfomycin resistance (PMID: 28851843). I wonder whether this gene is inherent and whether the name *fosA11* is proper.

>>**RESPONSE:** Thanks for the reviewer's insightful comments. This is a very important point regarding the nomenclature of *fosA* gene and gene transferable properties. The drug resistance gene was named according to the order of the latest reported resistance gene *fosA10* (PMID: 32431524), and several previously reported *fosA* genes present on the chromosome were also named by this method. Such as *fosA2* located in chromosomes of *Enterobacter cloacae* (PMID: 21392044) and *fosA7* founded in chromosomes of *Salmonella* (PMID: 28533247). The gene is indeed located on the chromosome of *Providencia*, and the current sequencing analysis found that no known mobile elements were found in the upstream and downstream of the gene; moreover, search throughout the NCBI database and the analysis of *Providencia* collected in our laboratory it can be seen that this gene only exists on the chromosomes of a small part of *Providencia*, and most *Providencia* have not found this gene, so it can be called inherent drug resistance gene in small part of *Providencia*.

>>**COMMENT:** The minor comment is that the Table 1 could be transferred to supplement materials.

>>**RESPONSE:** Thanks for the reviewer's insightful comments. the Table 1 has been transferred to supplement materials and merged with Table S1. (See Supplement Tables S1)

Referee: 2 (Comments to the Author):

>>**COMMENT:** In this work, Lu et al provide novel insights into fosfomycin resistance in *Providencia rettgeri*. Specifically, using a UTI *P. rettgeri* isolate, *P. rettgeri* W986, the authors present work on a novel fos resistance gene, which they name *fosA11*. They provide evidence of catalytic activity against fosfomycin, although FosA11's affinity for fosfomycin is lower than the characterized FosA3 from *E. coli*. This study reports that this novel gene, *fosA11* is chromosomally-encoded and prevalent among *Providencia* species, which could explain intrinsic resistance to fosfomycin in such isolates.

Over all the study is linearly performed and provides new insight into the field. However, there are areas that lack rigor, which need to be addressed.

>>**COMMENT:** 1) There are no statistical analyses for the MICs and more extensive description of how experiments were performed would be useful.

>>**RESPONSE:** Thanks for the reviewer's careful reading and constructive suggestions, which has significantly improved the presentation of our manuscript. (See Lines 173-176 and Lines303-306)

>>**COMMENT:** 2) Critical information shown in the figures and tables is oftentimes not discussed in the main text. For example, it is not mentioned in the text that G6P is not required for uptake of fosfomycin by *Providencia*. However, this is demonstrated in Table 3. Does *Providencia* not have the Uhp system described in *E. coli*? This would be good to discuss, given that *Providencia* is understudied and the authors have performed whole genome sequencing.

>>**RESPONSE:** Thanks for the reviewer's meaningful comments. We have added discussion contents of the table to the main text. (See Lines 308-314)

>>**COMMENT:** 3) There is no description of how the candidate fosfomycin resistance genes were identified (Lines297-300). As mentioned above, it would be more useful to provide a table of the genes that are putatively associated with

fosfomycin or other drug resistance in the main text rather than the supplement.

>>**RESPONSE:** Thanks for the reviewer's constructive suggestions. We agree with the reviewer's suggestion that we have described in more detail on the process of how potential fosfomycin resistance genes were discovered. (See Lines 303-307) Moreover, we have transferred the table of the genes that are putatively associated with fosfomycin or other drug resistance from the supplementary material to the main text. (See Table 5)

>>**COMMENT:** 4) There should be some genetic validation with mutagenesis of the catalytic pocket of FosA11 to validate that this mutant stops imparting fosfomycin resistance to *E. coli*.

>>**RESPONSE:** Thanks for the reviewer's insightful comments. based on reviewer's suggestion we have completed the mutagenesis experiments and according to the experiment results, the corresponding part of the papper has been improved. (See Lines 181-182 and Lines 369-392 and Table 1)

>>**COMMENT:** 5) The authors state that there are numerous *Providencia* species that have the *fosA11*. Since the authors have run the analyses, they should be quantitative here: "out of xx sequenced genomes, Y number of genomes code for *fosA11*".

>>**RESPONSE:** Thanks for the reviewer's valuable comments. We have revised the detailed expression of the text on analyses results of numerous *Providencia* species that have the *fosA11* according to reviewer's detailed suggestion. (See Lines 328-329)

Minor

>>**COMMENT:** The article needs editing for English. In some cases, like in lines 238-241 it is hard to understand what the authors meant to state.

>>**RESPONSE:** Thanks for the reviewer's useful advice. The revised paper has been sent to professional English staff for language polishing.

>>**COMMENT:** Please make sure that the references are separated by a space between the reference and the last word in a sentence.

>>**RESPONSE:** Thanks for the reviewer's careful advice. We have revised the citation format throughout the manuscript according to reviewer's detailed suggestion.

>>**COMMENT:** The MICs for the other drugs for strain W986 should be given in the form of a table instead of described in the text. (lines 241-248)

>>**RESPONSE:** Thanks for the reviewer's pertinent advice. We've made the appropriate changes to the paper. (See Supplement Tables S4 and Lines 250-253)

>>**COMMENT:** In materials and methods, did you mean 10 micromolar or 10 millimolar IPTG?

>>**RESPONSE:** Thanks for the reviewer's insightful comments. It is 10 millimolar, and has made corresponding changes in the paper. (See Lines 201)

Again, we would like to extend our gratitude to you and the reviewers for bringing these issues to our attention. My co-authors and I agree that the changes made to the manuscript have greatly improved this manuscript. We really appreciate it.

Regards,

Shenghai Wu

wu_shai@163.com

Re: Spectrum02542-23R1 (*fosA11*, a Novel Chromosomal Encoded Fosfomycin Resistance Gene Identified in *Providencia rettgeri*)

Dear Mr. Shenghai Wu:

Your manuscript has been accepted, and I am forwarding it to the ASM production staff for publication. Your paper will first be checked to make sure all elements meet the technical requirements. ASM staff will contact you if anything needs to be revised before copyediting and production can begin. Otherwise, you will be notified when your proofs are ready to be viewed.

Sincerely,
Silvia Cardona
Editor
Microbiology Spectrum

Reviewer #2 (Comments for the Author):

The authors have adequately addressed all reviewer comments.
There is a minor typo in line 334: Is MIV meant to be MIC? Please check.